applied mathematics/mathematical modelling/ health and disease and epidemiology

COVID-19, SARS-CoV-2, epidemic model, vaccination, disease modelling

**Author for correspondence:**
Fernando Saldaña
e-mail: fernando.saldana@im.unam.mx

# The trade-off between mobility and vaccination for COVID-19 control: a metapopulation modelling approach

## Fernando Saldaña and Jorge X. Velasco-Hernández

Instituto de Matemáticas, Campus Juriquilla, 76230, Universidad Nacional Autónoma de México, Querétaro, Mexico

FS, 0000-0003-0558-1169

November 2020 received a string of encouraging results from leading vaccine developers raising hopes for the imminent availability of an effective and safe vaccine against the SARS-CoV-2. In the present work, we discuss the theoretical impact of introducing a vaccine across a range of scenarios. In particular, we investigate how vaccination coverage, efficacy and delivery time affect the control of the transmission dynamics in comparison to mobility restrictions. The analysis is based on a metapopulation epidemic model structured by risk. We perform a global sensitivity analysis using the Sobol method. Our analysis suggest that the reduction of mobility among patches plays a significant role in the mitigation of the disease close to the effect of immunization coverage of 30% achieved in four months. Moreover, for an immunization coverage between 20% and 50% achieved in the first half of 2021 with a vaccine efficacy between 70% and 95%, the percentage reduction in the total number of SARS-CoV-2 infections is between 30% and 50% by the end of 2021 in comparison with the no vaccination scenario.

## 1. Introduction

Even though several countries have implemented social distancing and other non-pharmaceutical interventions to mitigate the SARS-CoV-2 epidemic, the number of infections continues to increase around the world [1]. After 11 months SARS-CoV-2 has infected more than 60 million and killed more than 1.4 million people globally.

According to the Johns Hopkins University database, Latin America has been one of the most affected areas with more than

430 000 deaths registered, which represents approximately 30% of the worldwide total. In particular, Mexico's death rate as a percentage of all confirmed cases is around 10%, being one of the highest in the world as of 20 November 2020.

In response to the COVID-19 crisis, there has been an unprecedented collaborative effort by researchers to develop an effective and safe vaccine. There are currently more than 100 COVID-19 vaccine candidates under development, with a number of these in the human trial phase [2]. At the time of writing (November 2020), there has been a string of encouraging results from leading vaccine developers. On 9 November, the drug company Pfizer announced that their vaccine (co-developed by BioNTech) was more than 90% effective at preventing COVID-19 with no serious safety concerns [3]. Two days later, the Russian developers behind the Sputnik V vaccine announced, in a press release, that their vaccine was 92% effective [4]. Pfizer & BioNTech results are based on 94 cases, whereas Sputnik V's efficacy was measured with 20 COVID-19 cases. On 16 November, the company Moderna announced that their vaccine is more than 94% effective at preventing COVID-19, based on an analysis of 95 cases [5]. Finally, AstraZeneca and the University of Oxford announced on 23 November that their vaccine has on average 70% effectiveness based on data from late-stage clinical trials. Pfizer & BioNTech and Moderna's vaccines can only be kept outside of ultracold freezers for a few weeks, making their storage and distribution very challenging in regions with poor infrastructure [6]. However, unlike the Moderna and Pfizer's immunizations, the AstraZeneca-Oxford vaccine does not require a sophisticated cold chain to stay in good condition and is therefore expected to be easier to distribute across all countries.

As vaccine development continues, there is an urgent need to assess the population-level impact of vaccine introduction [7–10]. The main goal of this work is to investigate the theoretical impact of introducing a vaccine under a number of scenarios. Rather than obtaining quantitative predictions of the epidemic course, we explore a spectrum of possibilities to gain qualitative insight on how much the initial introduction of the vaccine can slow the spread of the disease. Many things remain unknown: how effective vaccines will be, the duration of vaccine-induced immunity, and vaccine supply. We particularly concentrate on the impact of vaccine efficacy, coverage, duration of protection and delivery time on the transmission dynamics of COVID-19. Our analysis is based on a risk-structured metapopulation epidemic model that incorporates mobility following [11]. The risk of acquiring an infection is quantified as a function of the time spent within a particular location where the effective number of contacts that an individual has varies across locations, being higher in places with some form of mass gathering, e.g. public transportation, malls, religious ceremonies, universities, etc. [12]. We exemplify our model with the case of Mexico City.

The paper is structured as follows. In the next section, we formulate our mathematical model and obtain the basic and effective reproduction numbers. In §3, we investigate a range of vaccination scenarios that can help to evaluate the population-level impact of a COVID-19 vaccine in a Mexican setting, and in §4 we analyse the role of mobility and patch-dependent risk in the proposed vaccination scenarios. To have a more complete understanding of how vaccine-associated parameters affect the model outcomes, in §5 we perform a variance-based global sensitivity analysis (GSA) using the Sobol method. Finally, in §6, we summarize our findings and discuss future work.

# 2. Material and methods

## 2.1. Model formulation

We present a risk-structured metapopulation epidemic model based on an extension of the classical Kermack–McKendrick SEIR model with the approach of Bichara *et al.* [11]. In this context, metapopulation epidemic models assume that demographic and disease dynamics occur at a comparable time scale, but individuals' movement among patches occurs at a faster time scale.

For our model formulation, we consider a three-patch geographically structured population. Each of the these three patches, labelled $P1$, $P2$ and $P3$, represents risk areas with low, middle and high risk of infection, respectively. The risk of infection may depend on environmental, socioeconomic, demographic and sanitary conditions [12]. The resident population $N_i$ of patch $i$, which consists of individuals who normally live in that area, is further subdivided into eight mutually exclusive compartments of non-vaccinated susceptible ($S_i$), vaccinated susceptible ($V_i$), exposed ($E_i$), vaccinated exposed ($\tilde{E}_i$), asymptomatic infectious ($A_i$), symptomatic infectious ($I_i$), reported infectious ($I_i^r$) and recovered ($R_i$), hence $N_i = S_i + V_i + E_i + \tilde{E}_i + A_i + I_i + I_i^r + R_i$ with $i = 1, 2, 3$.

Movement among patches is described using the residence time matrix $\mathbf{P} = [p_{ij}]$, $i, j = 1, 2, 3$, where $p_{ij}$ represents the fraction of total daily time that $i$-residents spend in $j$-environments, with $\sum_{j=1}^{3} p_{ij} = 1$ for all $i$. $N_j^e$ is the effective population of patch $j$ at time $t$, that is, the number of individuals who are physically present in patch $j$ at time $t$, then

$$N_j^e = \sum_{k=1}^{3} p_{kj} N_k. \tag{2.1}$$

This notation is also used for the disease compartments, for example, $I_j^e = \sum_{k=1}^{3} p_{kj} I_k$ is the effective infectious population in patch $j$ at time $t$.

The model is given by the following system of differential equations:

$$
\left.
\begin{aligned}
\frac{\mathrm{d}S_i}{\mathrm{d}t} &= b_i N_i - \sum_{j=1}^{3} \beta_j p_{ij} S_i \frac{I_j^e + \alpha_j A_j^e}{N_j^e} - u_i S_i + w_i R_i + \theta_i V_i - d_i S_i, \\
\frac{\mathrm{d}V_i}{\mathrm{d}t} &= u_i S_i - \sum_{j=1}^{3} (1 - \psi_i) \beta_j p_{ij} V_i \frac{I_j^e + \alpha_j A_j^e}{N_j^e} - \theta_i V_i - d_i V_i, \\
\frac{\mathrm{d}E_i}{\mathrm{d}t} &= \sum_{j=1}^{3} \beta_j p_{ij} S_i \frac{I_j^e + \alpha_j A_j^e}{N_j^e} - k_i E_i - d_i E_i, \\
\frac{\mathrm{d}\tilde{E}_i}{\mathrm{d}t} &= \sum_{j=1}^{3} \beta_j p_{ij} ((1 - \psi_i) V_i) \frac{I_j^e + \alpha_j A_j^e}{N_j^e} - k_i \tilde{E}_i - d_i \tilde{E}_i, \\
\frac{\mathrm{d}A_i}{\mathrm{d}t} &= (1 - \rho_i) k_i E_i + (1 - \tilde{\rho}_i) k_i \tilde{E}_i - \gamma_i^a A_i - d_i A_i, \\
\frac{\mathrm{d}I_i}{\mathrm{d}t} &= \rho_i k_i E_i + \tilde{\rho}_i k_i \tilde{E}_i - \gamma_i I_i - \nu_i I_i - d_i I_i, \\
\frac{\mathrm{d}I_i^r}{\mathrm{d}t} &= \nu_i I_i - \gamma_i^r I_i^r - d_i I_i^r \\
\frac{\mathrm{d}R_i}{\mathrm{d}t} &= \gamma_i^a A_i + \gamma_i I_i + \gamma_i^r I_i^r - w_i R_i - d_i R_i, \quad i = 1, 2, 3.
\end{aligned}
\right\} \tag{2.2}
$$

and

The parameters $b_i$ and $d_i$ represent the *per capita* birth and death rates in patch $i$, respectively. Susceptible individuals are vaccinated at a rate $u_i$ and acquire the infection after an effective contact with a symptomatic infectious person with an effective contact rate $\beta_j$ that is an index of the patch-specific risk ($j = 1, 2, 3$). The parameter $0 < \alpha_i < 1$ measures the relative infectiousness of the asymptomatic infectious class in relation to symptomatic individuals in patch $i$ ($i = 1, 2, 3$). The vaccine reduces the force of infection (FOI) by a factor $\psi_i$ with $0 < \psi_i < 1$. The parameter $1/k_i$ represents the mean incubation period; after this, a fraction $\rho_i$ of the exposed class $E_i$ transition to the symptomatic infectious class $I_i$, while the other fraction $1 - \rho_i$ enter the asymptomatic infectious class $A_i$. The parameter $\tilde{\rho}_i$ represents the symptomatic fraction in the vaccinated exposed class $\tilde{E}_i$ ($i = 1, 2, 3$) with $\tilde{\rho} \le \rho$. Individuals in the symptomatic infectious class are reported at a rate $\nu_i$ and are effectively isolated and they no longer contribute to the FOI. The parameters $\gamma_i$, $\gamma_i^a$ and $\gamma_i^r$ are the recovery rates for classes $I_i$, $A_i$ and $I_i^r$, respectively ($i = 1, 2, 3$).

Note that in our model, vaccination not only may prevent SARS-CoV-2 infection but also it may prevent the symptomatic disease COVID-19. Moreover, unlike previous COVID-19 epidemic models (e.g. [9,13–17]) system (2.2) allows the possibility of reinfections. This is essential since it is not yet known how long natural immunity will last and there have been already confirmed cases of coronavirus reinfection [18]. Furthermore, it is also uncertain whether vaccine-induced immunity will be short- or long-lived; therefore, we assume natural and vaccine-induced immunity are lost at rates, $w_i$ and $\theta_i$, respectively ($i = 1, 2, 3$).

Since we look at the dynamics of model (2.2) over a relatively short time interval, we assume a constant population size with the birth rate equal to the death rate, that is, $b_i = d_i$. Therefore, the resident population in patch $i$ is a constant $N_i^*$ ($i = 1, 2, 3$). The boundedness, positiveness and continuity of the solutions are fairly straightforward to obtain from the model equations and the fixed population size assumption. Hence, system (2.2) is mathematically and epidemiologically well posed [19].

Note that patch $i$ always has a disease-free equilibrium of the form

$$E_0^i = (S_i^*, V_i^*, E_i^*, \tilde{E}_i^*, A_i^*, I_i^*, I_i^{r*}, R_i^*) = \left( \frac{(\theta_i + d_i) b_i N_i^*}{d_i (u_i + \theta_i + d_i)}, \; N_i^* - S_i^*, 0, 0, 0, 0, 0, 0 \right).$$

The three-patch metapopulation epidemic model (2.2) is at the disease-free equilibrium if every patch is at the disease-free equilibrium.

## 2.2. The basic reproduction number

In the simple case in which patches are isolated from the others, that is, $p_{ij} = \delta_{ij}$, where $\delta_{ij}$ is the Kronecker delta ($i, j = 1, 2, 3$); the patch-specific basic reproduction numbers are easily obtained

$$\mathcal{R}_0^i = (1 - \rho_i)T_A^i + \rho_i T_I^i, \quad i = 1, 2, 3, \tag{2.3}$$

where $T_A^i = \alpha_i \beta_i k_i / (k_i + d_i)(\gamma_i^a + d_i)$ and $T_I^i = \beta_i k_i / (k_i + v_i + d_i)(\gamma_i + d_i)$, measure the contributions of the asymptomatic and symptomatic infectious classes to the production of new infections, respectively. The patch-specific basic reproduction number is a threshold quantity and $\mathcal{R}_0^i < 1$ implies the local stability of $E_0^i$. However, we have to remark that this result is only local and there exists the possibility that the introduction of an imperfect vaccine in system (2.2) can lead to the emergence of a backward bifurcation [20].

To compute the basic reproduction number in the presence of movement, we take a next-generation approach [21] using the method of [22]. Ordering the infected subsystem ($I_i^r$ class can be omitted because they do not contribute to the FOI) as

$$E_1, A_1, I_1, E_2, A_2, I_2, E_3, A_3, I_3$$

we obtain a block matrix $\mathbf{F} = [\tilde{\mathbf{F}}_{ij}]$, where for $i, j = 1, 2, 3$, $\tilde{\mathbf{F}}_{ij}$ is an $3 \times 3$ matrix with

$$\tilde{\mathbf{F}}_{ij} = \begin{bmatrix} 0 & \lambda_{ij}^a & \lambda_{ij} \\ 0 & 0 & 0 \\ 0 & 0 & 0 \end{bmatrix}, \quad \lambda_{ij}^a = \sum_{k=1}^{3} \alpha_k \beta_k \frac{p_{ik} p_{jk} N_k^*}{N_k^{e*}} \quad \text{and} \quad \lambda_{ij} = \sum_{k=1}^{3} \beta_k \frac{p_{ik} p_{jk} N_k^*}{N_k^{e*}}. \tag{2.4}$$

The matrix $\mathbf{V}$ is a block diagonal matrix $\mathbf{V} = \mathrm{diag}(\tilde{\mathbf{V}}_{ii})$, where for $i = 1, 2, 3$, $\tilde{\mathbf{V}}_{ii}$ is an $3 \times 3$ matrix with

$$\tilde{\mathbf{V}}_{ii} = \begin{bmatrix} k_i + d_i & 0 & 0 \\ -(1 - \rho_i)k_i & \gamma_i^a + d_i & 0 \\ -\rho_i k_i & 0 & \gamma_i + d_i \end{bmatrix}. \tag{2.5}$$

The next-generation matrix is the block matrix $\mathbf{K} = \mathbf{F}\mathbf{V}^{-1} = [\tilde{\mathbf{F}}_{ij}\tilde{\mathbf{V}}_{jj}^{-1}]$, $i, j = 1, 2, 3$, and the basic reproduction number in the presence of movement is $\mathcal{R}_0 = \rho(\mathbf{K})$, where $\rho(\cdot)$ is the spectral radius. Note that by definition $\mathcal{R}_0$ assumes a fully susceptible population and, hence, control measures, such as mass vaccination, that reduce the the number of susceptible individuals in the population technically do not affect the value of $\mathcal{R}_0$ [23].

The same method allows us to obtain the effective reproduction number $\mathcal{R}_e$ as the spectral radius of the controlled next-generation matrix $\mathbf{K}_c = \mathbf{F}_c \mathbf{V}_c^{-1}$, where $\mathbf{F}_c = [\tilde{\mathbf{F}}_{ij}^c]$, where for $i, j = 1, 2, 3$, $\tilde{\mathbf{F}}_{ij}^c$ is an $4 \times 4$ matrix with

$$\tilde{\mathbf{F}}_{ij}^c = \begin{bmatrix} 0 & 0 & \varphi_{ij}^a & \varphi_{ij} \\ 0 & 0 & \chi_{ij}^a & \chi_{ij} \\ 0 & 0 & 0 & 0 \\ 0 & 0 & 0 & 0 \end{bmatrix},$$

$$\varphi_{ij}^a = \sum_{k=1}^{3} \alpha_k \beta_k \frac{p_{ik} p_{jk} S_k^*}{N_k^{e*}}, \quad \varphi_{ij} = \sum_{k=1}^{3} \beta_k \frac{p_{ik} p_{jk} S_k^*}{N_k^{e*}},$$

and

$$\chi_{ij}^a = \sum_{k=1}^{3} \alpha_k \beta_k (1 - \psi_k) \frac{p_{ik} p_{jk} V_k^*}{N_k^{e*}}, \quad \chi_{ij} = \sum_{k=1}^{3} \beta_k (1 - \psi_k) \frac{p_{ik} p_{jk} V_k^*}{N_k^{e*}}.$$

The matrix $\mathbf{V}_c$ is a block diagonal matrix $\mathbf{V}_c = \mathrm{diag}(\tilde{\mathbf{V}}_{ii}^c)$, where for $i = 1, 2, 3$, $\tilde{\mathbf{V}}_{ii}^c$ is an $4 \times 4$ matrix with

$$\tilde{\mathbf{V}}_{ii}^c = \begin{bmatrix} k_i + d_i & 0 & 0 & 0 \\ 0 & k_i + d_i & 0 & 0 \\ -(1 - \rho_i)k_i & -(1 - \tilde{\rho}_i)k_i & \gamma_i^a + d_i & 0 & 0 \\ -\rho_i k_i & -\tilde{\rho}_i k_i & 0 & \gamma_i + d_i \end{bmatrix}. \tag{2.6}$$

## 2.3. Model parameters

We retrieved the baseline values for some of our model parameters using COVID-19 epidemic data from the Mexican Federal Health Secretary [24] (Secretaría de Salud Mexico) and estimations from previous studies with data from Mexico City [14,25,26]. Evidence suggests that approximately four in five people infected with SARS-CoV-2 develop symptoms [27], therefore, we set $\rho_i = 0.8$. The incubation period for COVID-19 ranges from 2 to 14 days with an average between 5 and 7 days, we choose the estimation $1/k_i = 5.99$ days from [25]. The mean recovery rates for the asymptomatic, symptomatic and reported infectious classes are $\gamma_i^a = 1/14$, $\gamma_i = 1/10.81$ and $\gamma_i^r = 1/5.0$ d$^{-1}$, respectively. The average progression rate from the symptomatic-infectious class to the reported infectious class is $v_i = 1/3.0$ d$^{-1}$ [14]. The relative infectiousness of the asymptomatic infectious class in relation to symptomatic individuals is $\alpha_i = 0.45$ [28]. The life expectancy of the Mexican population is roughly 70 years; hence, $d_i = 1/(70 * 365)$ d$^{-1}$. The results presented in [18] suggest that reinfections are more likely to occur at 12 months after infection. For this study, we assume natural immunity lasts a year, i.e. $1/w_i = 365$ days unless otherwise stated. These epidemiological parameters are common for each of the three patches $P1$, $P2$, $P3$, since they are geographical regions that comprise people of all ages. The value of the basic reproduction number for COVID-19 in Mexico has been estimated to be between 2.95 and 5.89 with a mean value of 3.87 [25]. Using these estimations, we obtain the following values for the transmission coefficients: $\beta_1 = 0.31$, $\beta_2 = 0.40$ and $\beta_3 = 0.53$. The initial time for our study is 20 November 2020, and the time horizon to study the effects of vaccination is 365 days. The initial conditions are fixed according to the epidemiological data on 20 November [24]. Since the risk distribution is constantly changing as measured by the number of active cases per geographical area, we simplify assuming that low-risk areas are more frequent than high-risk areas. In particular, we assume a distribution of 50% in $P1$, 30% in $P2$ and 20% in $P3$.

# 3. COVID-19 vaccination scenarios

In this section, we investigate several vaccination scenarios to evaluate the population-level impact of a COVID-19 vaccine. Each scenario is based on a set of numerical values for the vaccine-associated parameters $\Theta = \{(\tilde{\rho}_i, \psi_i, \theta_i, C_i, T_i), \ i = 1, 2, 3\}$ where $C_i$ is the target immunization coverage and $T_i$ the target time to achieve that coverage. The vaccination rate $u_i$ is obtained from the approximation $1 - \exp(-u_i T_i) = C_i$ [29]. To formulate our scenarios, we take into consideration the currently available information of the COVID-19 vaccine candidates [2]. The expected vaccine efficacy is between 70% and 95% and the duration of vaccine-induced immunity should be at least six months. There is still no reliable information on the proportion of symptomatic infections in vaccinated individuals so for simplicity, we assume this proportion is less than 50%. Besides, the Mexican public health authorities expect to achieve immunization coverage between 10% and 50% in the year 2021 [30]. Under these considerations, we propose three scenarios (table 1) attempting to reflect a range of possibilities between worst-case and optimistic conditions. Each scenario is further subdivided according to the time needed to reach the target immunization coverage $T_i$: (a) one month, (b) three months and (c) five months.

Moreover, considering that it is not yet known how long natural immunity might last and how common reinfection is [18]; we investigate each of the vaccination scenarios for a duration of protective immunity of 180 and 365 days. For all cases, the introduction of the vaccine starts on 1 January 2021. Parameter values that vary among the vaccination scenarios are listed in table 1, while parameter values common to all scenarios are described in §2.3. We start our simulations in the simple case in which individuals spend the same time in all patches; hence, the residence time matrix is $\mathbf{P}_{1/3} = [p_{ij}]$ with $p_{ij} = 1/3$ for all $i, j$. For this mobility matrix and the baseline parameter values, the value of the basic reproduction number (which does not consider vaccination) is $\mathcal{R}_0 = 1.29$.

Figure 1 shows the cumulative number of reported cases per day for the proposed vaccination scenarios in table 1 according to the time needed to achieve the target vaccination coverage: (a) one month, (b) three months and (c) five months. The no vaccination case is shown in red and the data correspond to the official cumulative confirmed cases in Mexico City until 20 November 2020. The assumed duration of natural immunity is 180 days, therefore, the number of reported cases in the no vaccination scenario increases rapidly reaching more than 500 000 cases by the end of 2021. In the most optimistic scenario, Scenario 3 (a), the introduction of the vaccine allows maintaining the cumulative reported cases around 300 000 cases by the end of 2021, achieving more than 40% reduction in the reported cases. For Scenario 3 (a), the effective reproduction number is $\mathcal{R}_e = 0.59$ so vaccination decreases significantly (more than 50%) the value of $\mathcal{R}_0$. Figure 2 also shows the cumulative number of reported cases. Yet, in this case, the assumed

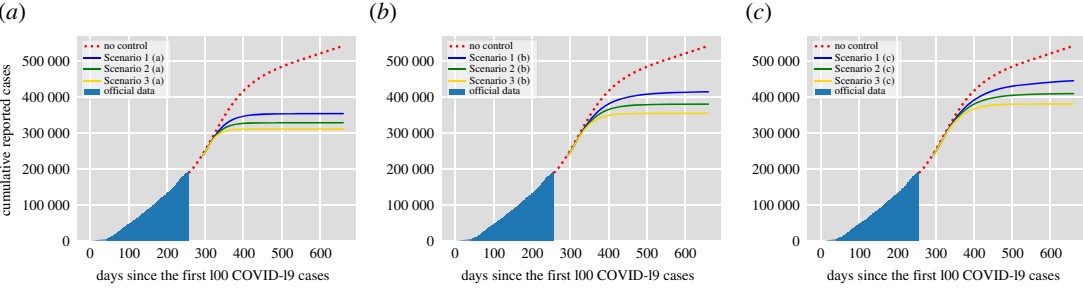

**Figure 1.** Cumulative number of reported cases per day for the vaccination scenarios in table 1 according to the time needed to achieve the target vaccination coverage: (*a*) one month, (*b*) three months and (*c*) five months. The data correspond to the official cumulative confirmed cases in Mexico City until 20 November 2020. The assumed duration of natural immunity is 180 days.

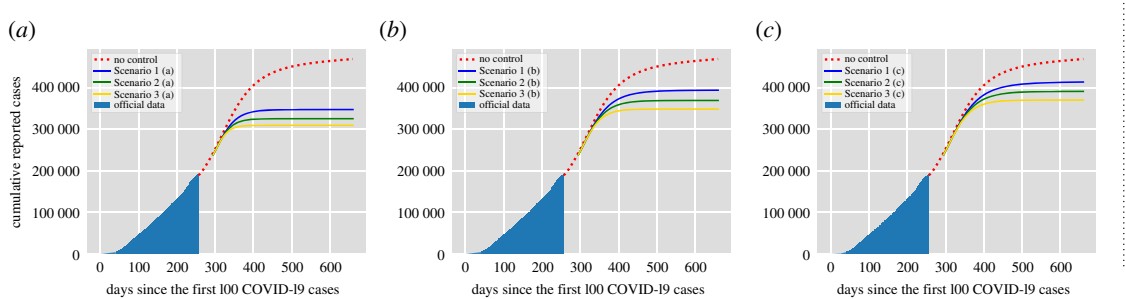

**Figure 2.** Cumulative number of reported cases per day for the vaccination scenarios in table 1 according to the time needed to achieve the target vaccination coverage: (*a*) one month, (*b*) three months and (*c*) five months. The data correspond to the official cumulative confirmed cases in Mexico City until 20 November 2020. The assumed duration of natural immunity is 365 days.

**Table 1.** Description of the parameter values for the three COVID-19 vaccination scenarios. Each vaccination scenario is further subdivided according to the time needed to reach the target vaccination coverage: (*a*) one month, (*b*) three months and (*c*) five months. The parameters are the same for all patches ($i = 1, 2, 3$) and the vaccination rate $u_i$ is obtained from the approximation $1 - \exp(-u_i T_i) = C_i$, where $C_i$ is the target coverage and $T_i$ the time wished to achieve coverage $C_i$.

| parameter | range | Scenario 1 | Scenario 2 | Scenario 3 |
|---|---|---|---|---|
| $C_i$ | [10%, 50%] | 30% | 40% | 50% |
| $\psi_i$ | [70%, 95%] | 70% | 80% | 90% |
| $\tilde{\rho}_i$ | [0%, 50%] | 50% | 30% | 10% |
| $\theta_i^{-1}$ | [180, 365] days | 180 | 250 | 365 |

$T_i \in [1, 5]$ months: (*a*) one month (*b*) three months (*c*) five months.

duration of natural immunity is 365 days and, hence, in the no control case, the reported cases by the end of 2021 is around 420 000 cases. From figure 2, we see that if the duration of natural immunity is one year, the introduction of a vaccine with approximately 90% effectiveness (Scenario 3) allows controlling the infection before the end of 2021. However, this is not the case if natural immunity lasts only half a year (see, for example, Scenario 1 (*c*) in figure 1). For all the scenarios explored, the value of the effective reproduction number ranges between 0.59 and 1.13. Hence the effect of vaccination on the reduction of the $\mathcal{R}_0$ values varies significantly for different conditions. However, we have to remark that we did not consider the effect of non-pharmaceutical interventions which play a significant role in the control of $\mathcal{R}_e$.

# 4. Residence times and patch-dependent risk

## 4.1. The role of mobility

Here, we investigate the role of mobility and patch-dependent risk in the proposed vaccination scenarios. The description of movement among regions is a matter of particular interest for the study of infectious

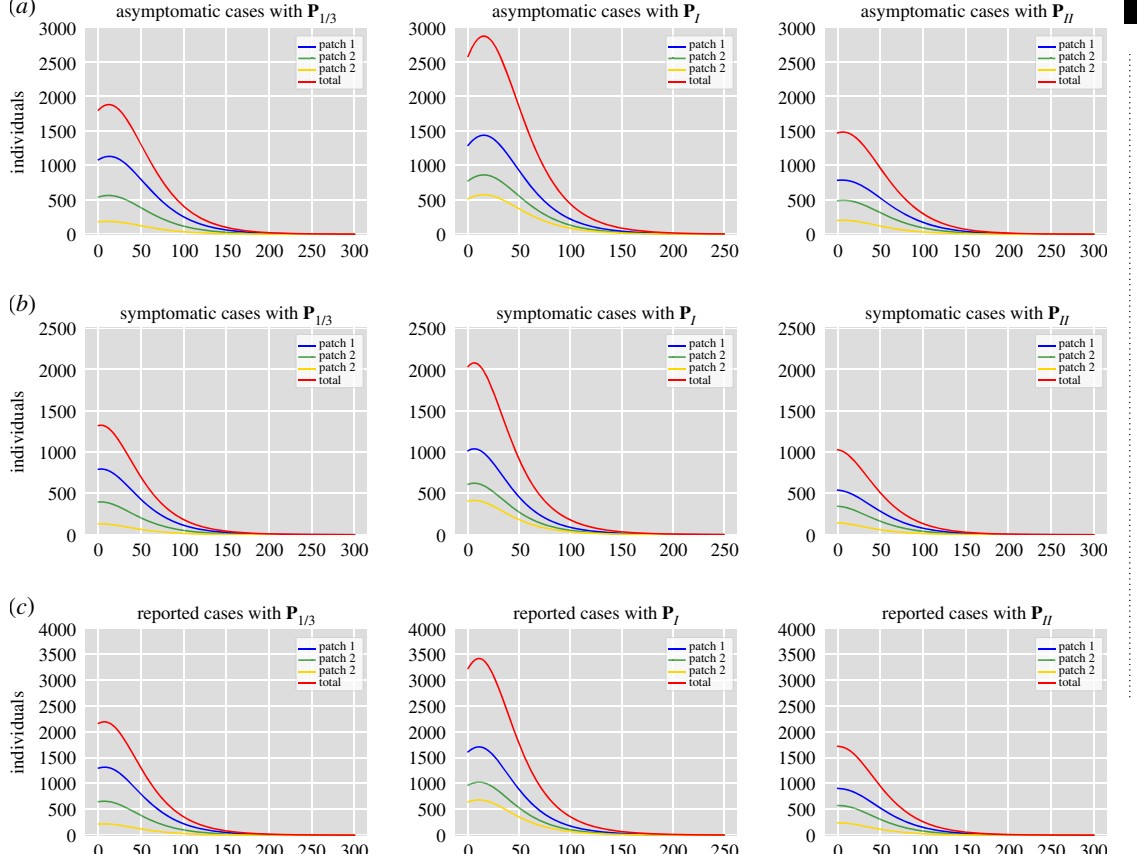

**Figure 3.** (a) Asymptomatic cases per patch. (b) Symptomatic cases per patch. (c) Reported cases per patch. Left column: The mobility matrix is $\mathbf{P}_{1/3}$. Middle column: The mobility matrix is $\mathbf{P}_I$. Right column: The mobility matrix is $\mathbf{P}_{II}$. For all cases, the parameter values are fixed according to Scenario 2 (b) in table 1. The simulations start on 1 January 2021, with an initial condition obtained from the results of model (2.2) simulated from 20 November 2020 until 31 December 2020, depending on the residence time matrix for each case. The assumed duration of natural immunity is 365 days.

disease transmission among spatially distinct populations and some studies had described the movement between two patches by relationships of the form $p_{ij} \propto N_i C_j N_j f(d)$, where $N_i$ is the respective resident population in patch $i$, $C_i$ is the population who work in patch $i$ (regardless of their resident patch) and $f(d)$ is a distance kernel [31].

System (2.2) models constant human mobility through the resident time matrix $\mathbf{P}$ and does not capture behavioural responses to disease dynamics that may optimize an index of well-being. Although this effect can be captured placing appropriate restrictions on the entries of $\mathbf{P}$ [12], we limit our study to the case where there is no behavioural change but we propose two mobility matrices (in addition to $\mathbf{P}_{1/3}$) that may be useful to support public health preparedness. (I) Individuals spend a considerable amount of time in high-risk areas. The residence time matrix for this case is $\mathbf{P}_I = [p_{ij}]$ with $p_{i3} = 0.7$ for all $i$, and $p_{ij} = 0.15$ with $i = 1, 2, 3, j = 1, 2$. (II) There is reduced mobility and individuals stay in their home patch most of their time. The residence time matrix for this case is $\mathbf{P}_{II} = [p_{ij}]$ with $p_{ij} = 0.8$ with $i = j$, and $p_{ij} = 0.1$ with $i \neq j$. All the results are summarized in the plots appearing in figure 3. The parameter values are fixed according to Scenario 2 (b) in table 1 which represents an intermediate case between worst-case and optimistic conditions.

The left column in figure 3 shows the number of individuals in the infected classes per patch for the residence time matrix $\mathbf{P}_{1/3}$, whereas the middle column shows the results for $\mathbf{P}_I$, and the left column the results for $\mathbf{P}_{II}$. Observe that the prevalence of the infection is, on average, higher for the residence time matrix $\mathbf{P}_I$ for all the infected classes in comparison with the other mobility matrices. Moreover, the residence time matrices also have an important influence on the reproduction numbers. The patch-specific reproduction numbers (which do not consider movement) for the parameter values chosen are $\mathcal{R}_0^1 = 1.02$, $\mathcal{R}_0^2 = 1.32$ and $\mathcal{R}_0^3 = 1.75$. In the presence of movement, the basic and effective reproduction numbers are $\mathcal{R}_0 = 1.12$ and $\mathcal{R}_e = 0.68$, $\mathcal{R}_0 = 1.5$ and $\mathcal{R}_e = 0.91$, $\mathcal{R}_0 = 1.11$ and $\mathcal{R}_e = 0.67$,

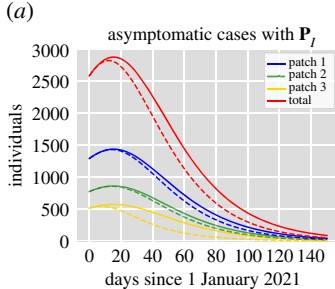
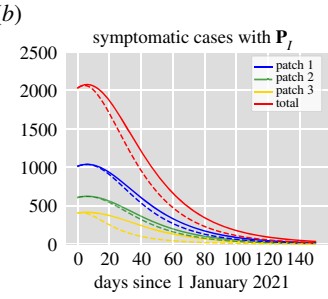
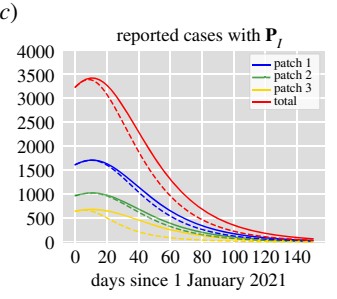

**Figure 4.** Dynamics of the infected classes per patch for equal vaccination rates (solid lines) in all patches and vaccine prioritization (dotted lines) for the high-risk patch $P3$. ($a$) Asymptomatic classes $A_1$ (blue), $A_2$ (green), $A_3$ (yellow). ($b$) Symptomatic classes $I_1$ (blue), $I_2$ (green), $I_3$ (yellow). ($c$) Reported classes $I_1^r$ (blue), $I_2^r$ (green), $I_3^r$ (yellow). The mobility matrix is $\mathbf{P}_I$.

for the matrices $\mathbf{P}_{1/3}$, $\mathbf{P}_I$ and $\mathbf{P}_{II}$, respectively. Therefore, if patches are strongly connected, the high-risk patch acts like a source patch promoting an increase in the transmission and the value of the reproduction numbers of the whole model. On the other hand, restricted mobility helps to maintain control of disease transmission. In particular, one can observe that the prevalence levels for the high mobility matrix $\mathbf{P}_I$ are almost twice the levels for the restricted mobility matrix $\mathbf{P}_{II}$. The disease is eradicated in all cases because the effective reproduction number, that depends on the successful deployment, coverage and efficacy of the vaccine, is less than unity.

## 4.2. Vaccination prioritization for high-risk areas

In previous simulations, we have considered equal vaccination rates for all patches. Nevertheless, long periods of residence in the high-risk patch promote an increase in transmission (figure 3). Hence, it is logical to expect that prioritizing the introduction of the vaccine in the high-risk patch can help mitigate the spread of the epidemic more effectively, at least when there is high mobility. Furthermore, if there is a limited vaccine supply, it may be expected that groups in the most affected areas may be recommended to get a COVID-19 vaccine first. What is more, despite the availability of the vaccine, there can be delays in acceptance or even refusal of vaccination in some groups of the population causing a low vaccine coverage in some areas. Taking this into consideration, we consider a scenario in which the high-risk patch has higher immunization coverage and faster deployment of the vaccine in comparison with the other two patches. For illustration purposes, we assume that public health authorities achieve 70% vaccination coverage after one month of the introduction of the vaccine in patch $P3$. We maintain patches $P1$ and $P2$ under the same conditions used in §4.1 (40% coverage in three months). We perform simulations to investigate if there is a reduction in the prevalence in comparison with the case of equal vaccination rates for the mobility matrix $\mathbf{P}_I$. The results depicted in figure 4 imply that prioritizing the high-risk patch reduces the transmission levels, especially in that patch. Nevertheless, this reduction appears to be very small in the other patches. This may be because the high-risk patch comprises less area than the other two patches and therefore high coverage in this patch is not enough to see a considerable reduction in the rest of the patches.

## 4.3. Coverage, efficacy and delivery time versus mobility

From the simulations shown in figure 4, one can see that even when prioritizing vaccination in the high-risk patch ($C_3 > C_1$, $C_2$ and $T_3 < T_1$, $T_2$) reduces the prevalence level, this reduction seems to be still less than the reduction achieved if there is restricted mobility (see the right column in figure 3). Therefore, we further explore how vaccination coverage, efficacy and delivery time affect the control of the transmission dynamics in comparison with the effect of mobility restrictions.

Figure 5 presents the results. For all plots, the total number of cumulative reported cases for the high-mobility case (the residence time matrix is $\mathbf{P}_I$) are presented by the red solid lines, and red dotted lines represent the low-mobility case (the residence time matrix is $\mathbf{P}_{II}$). These cumulative reported cases are obtained without vaccination and indicate that the expected number of reported infections for Mexico City is between 430 000 and 550 000 cases by the end of 2021 depending on the level of mobility. In figure 5$a$, we vary the vaccination coverage between 20% and 40% with fixed efficacy at 80% and delivery time in five months. In figure 5$b$, we vary vaccine efficacy between 70% and 90% with fixed

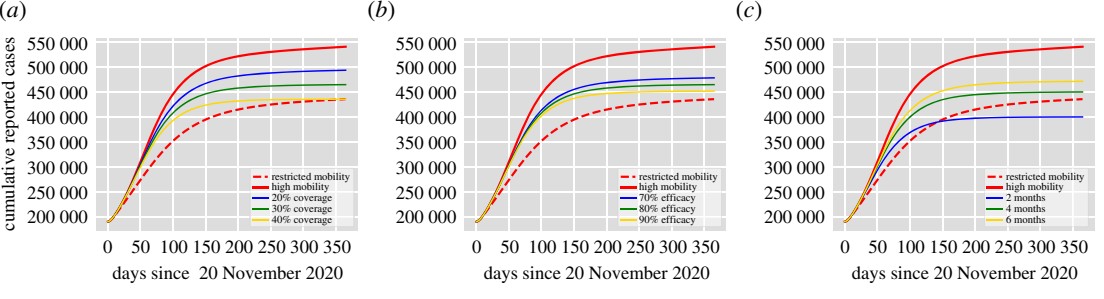

**Figure 5.** Impact of coverage, efficacy and delivery time on the cumulative number of reported cases in comparison with the impact of mobility. (*a*) Vaccination coverage between 20% and 40% with fixed efficacy at 80% and delivery time in five months. (*b*) Vaccine efficacy between 70% and 90% with fixed 30% coverage achieved in five months. (*c*) Delivery time between two to six months for fixed 80% efficacy and 30% coverage. The proposed scenarios are presented in solid lines (blue, green, yellow) and are obtained for the residence time matrix $\mathbf{P}_I$. For all sub-figures, the total number of cumulative reported cases for the residence time matrix $\mathbf{P}_I$ are presented by the red solid lines and red dotted lines represent the low-mobility case where the residence time matrix is $\mathbf{P}_{II}$.

30% coverage achieved in five months. In figure 5*c*, we vary delivery time between two and six months for fixed 80% efficacy and 30% coverage, respectively. The simulations (solid lines) are obtained for the high-mobility matrix $\mathbf{P}_I$. From figure 5*a*, we see that an efficacy of 80%, with 40% coverage achieved in five months has a similar effect in comparison with restricted mobility without vaccination. In figure 5*b*, one can see that even with 90% vaccine effectiveness if the coverage is 30% in five months, the reported cases are higher in comparison with the reduction achieved by restriction of mobility. On the other hand, if the 30% coverage is achieved very fast (two months) and vaccine effectiveness is 80%, the reduction in the number of cases is greater than the one obtained by restriction of mobility (figure 5*c*). In summary, the reduction of mobility among patches plays a significant role in the mitigation of the disease close to the effect of immunization coverage of 30% achieved in four months. Finally, note that for the ranges explored in figure 5, it seems that coverage and the time needed to achieved such coverage has more impact than vaccine efficacy in disease control.

## 5. Global sensitivity analysis

To have a better understanding of how vaccine-associated parameters affect model outcomes, we perform a variance-based GSA using the Sobol method [32]. In the presence of uncertainty in parameter values, GSA becomes an important methodology to quantify the sensitivity of model outcomes with respect to specific parameters as input factors. Within the Sobol framework, we estimate the first-order indices to measure the contribution by a single parameter alone and total-order indices that include the first-order effects but also all higher-order interactions. In this study, we investigate a range of vaccination scenarios varying coverage (20–60%), delivery time (one to five months), symptomatic fraction in the vaccinated-susceptible class (0–50%), duration of vaccine-induced immunity (6–12 months), and vaccine efficacy to prevent infection (70–95%). For the experiments, we varied the corresponding vaccine parameters within the proposed ranges for 5000 trials. During each trial, we randomly picked the value of each input parameter from the specified range and ran the model using those values. The outcomes of interest are the percentage reduction in SARS-CoV-2 cases in the asymptomatic, symptomatic and reported infected classes in each patch compared with the no vaccination case for the year 2021.

Results of the GSA are shown in figure 6 as histograms in which the $x$-axis corresponds to the percentage reduction in the corresponding infected class and the $y$-axis is the bin's frequency. The first row shows the percentage reduction in the asymptomatic classes, the middle row in the symptomatic classes and the bottom row in the reported classes. The histograms in blue, yellow and green correspond to the patches $P1$, $P2$ and $P3$, respectively. From the histograms, one can observe that the overall reduction for all patches and infected classes is close to 40% and the expected reduction in SARS-CoV-2 cases will be between 30% and 50% for the parameters proposed in this work.

Sobol's first and total indices are obtained for all the asymptomatic classes ($\sum_i A_i$) in figure 7*a*, the symptomatic classes ($\sum_i I_i$) in figure 7*b* and the reported classes ($\sum_i I_i^r$) in figure 7*c*. The $x$-axis corresponds to vector parameters $x = (x_1, x_2, x_3)$ with $x \in \{\theta, T, \psi, C, \tilde{\rho}\}$ and the vertical black lines represent 95% confidence intervals. The sensitivity analysis indicates that variations in vaccination

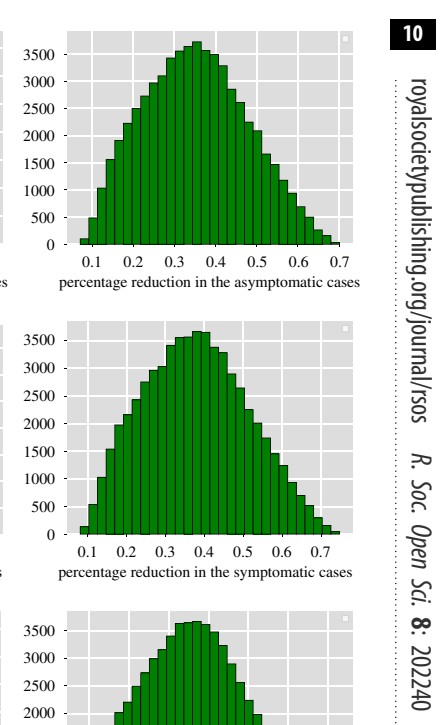

**Figure 6.** (*a*) Percentage reduction in the asymptomatic classes. (*b*) Percentage reduction in the symptomatic classes. (*c*) Percentage reduction in the reported classes. The histograms in blue, yellow and green correspond to the patches *P*1, *P*2 and *P*3, respectively. The *x*-axis corresponds to the percentage reduction in the corresponding infected class and the *y*-axis is the bin's frequency.

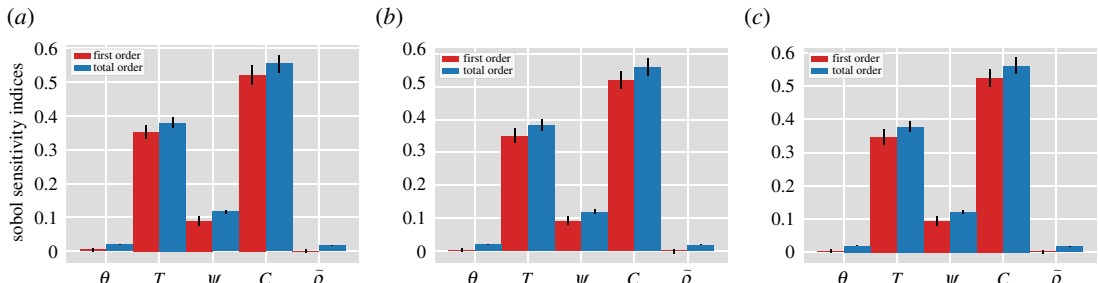

**Figure 7.** First and total order Sobol sensitivity indices with respect to the vaccine-associated parameters for the mean percentage reduction in: (*a*) the asymptomatic classes $\left(\sum_i A_i\right)$, (*b*) the symptomatic classes $\left(\sum_i I_i\right)$, and (*c*) and the reported classes $\left(\sum_i I_i^r\right)$. The *x*-axis corresponds to vector parameters $x = (x_1, x_2, x_3)$ with $x \in \{\theta, T, \psi, C, \tilde{\rho}\}$ and the vertical black lines represent 95% confidence intervals.

coverage contribute the most to the overall variance of reduction of the number of SARS-CoV-2 cases. Vaccine efficacy and the time needed to achieve the vaccination coverage also play a significant role in the model outcomes. The duration of vaccine-induced immunity and the reduction in symptomatic disease seems to have little impact on the model behaviour. This confirms the results obtained in §4. Finally, observe that the indices practically do not change among the infected classes.

# 6. Discussion

Less than a year after its emergence, the SARS-CoV-2 has taken more than one million lives and several countries are still struggling to maintain the number of infections under control. At the beginning of the

pandemic, in the absence of a vaccine, lockdowns, social distancing and other non-pharmaceutical interventions have helped to slow the spread of the virus. However, the prolonged implementation of lockdowns at a national level has caused economic and psychological distress for many citizens, especially in marginal groups and large regions in the developing world. The introduction of an effective and safe vaccine represents a powerful weapon to fight the pandemic and the hope for a return to normality. The encouraging results presented from leading vaccine developers in November 2020 are good news. However, the vaccine efficacy, protective time-span and coverage needed to obtain a quick and significant reduction in the number of COVID-19 infections may be difficult to achieve. Hence, to aptly manage expectations once the vaccines become available in the global market, it is important to analyse to what extent the initial deployment of the vaccine will help to control the spread of disease.

In this work, we investigate the theoretical population-level impact of introducing a vaccine across several scenarios of interest. The analysis is based on a three-patch risk-structured metapopulation epidemic model where patch infection prevalence depends on local environmental risk and interactions between connected patches. In each patch, the dynamics are governed by a Kermack–McKendrick-type model and the connection among patches is described by a residence-time matrix. Unlike several previous COVID-19 epidemic models, our system allows the possibility of reinfections and incorporates a vaccine that not only reduces susceptibility to infection but also prevents symptomatic disease. We explored how vaccination coverage, efficacy and delivery time affect the control of the transmission dynamics in comparison with the restriction of mobility. Our results show that an efficacy of 80%, with 40% coverage achieved in five months has a similar effect in comparison with restricted mobility without vaccination. The simulations also suggest that even with 90% vaccine effectiveness if the coverage is 30% in five months, the reported cases are higher in comparison with the reduction achieved by restriction of mobility. Concisely, the reduction of mobility among patches plays a significant role in the mitigation of the disease, close in performance to the effect of immunization coverage of 30% achieved in four months. Our model also presents some projections on the number of reported cases in Mexico City as a function of relevant model parameters and specific vaccine conditions. However, some limitations should be noted. First, we have not explicitly included a pre-symptomatic infectious class. Nevertheless, in Mexico and several other countries, there is still considerable uncertainty about the pre-symptomatic/asymptomatic carriers' contribution to the SARS-CoV-2 spread, see [33] and references therein. Hence, we consider that the asymptomatic class in our model is a rough but valid approximation for the contribution of pre-symptomatic/asymptomatic carriers. Second, most COVID-19 vaccines require two doses to achieve optimal protection but our model neglects the delay between doses administration. Moreover, parameter values are constantly adjusting as more data become available, and thus, these simulations are intended to explore plausible scenarios that can be of help for public health planning. They do not constitute quantitative predictions.

Considering the uncertainty associated with vaccine parameter values, we performed a GSA via Sobol's method. For our simulations, we varied the immunization coverage, delivery time, symptomatic fraction in the vaccinated-susceptible class, duration of vaccine-induced immunity and vaccine efficacy to prevent infection. Our results suggest that if public health authorities can achieve an immunization coverage between 20% and 50% in the first half of the year 2021 with a vaccine of effectiveness higher than 70%, the percentage reduction in the total number of SARS-CoV-2 infections in Mexico City is between 30% and 50% by the end of 2021 in comparison with the no vaccination scenario. Furthermore, if there is restricted mobility, the simulations suggest that for a vaccine efficacy of 90% (or higher) and coverage above 30% before June 2021, then a very low number of COVID-19 infections are expected by the end of 2021. The computational experiments of the sensitivity analysis also imply that variations on vaccination coverage and the time needed to achieve such coverage are the most important drivers in the reduction of the number of SARS-CoV-2 cases. Vaccine efficacy plays a significant role, whereas the duration of vaccine-induced immunity and the reduction in symptomatic disease seem to have little impact on the global model behaviour. As future work, we plan to extend our analysis to devise optimal vaccine allocation per age group to minimize deaths and symptomatic COVID-19 infections.

Data accessibility. Data and relevant code for this research work are stored in GitHub: https://github.com/fernandosaldanagarcia/SaldanaVelascoCovid19 and have been archived within the Zenodo repository: http://doi.org/10.5281/zenodo.4727829.

Authors' contributions. F.S. carried out the analysis of the model, numerical simulations and drafted the manuscript. J.X.V.-H. conceived and coordinated the study and helped draft the manuscript. Both authors interpreted the results.

Competing interests. We declare we have no competing interests.

Funding. We acknowledge support from DGAPA-PAPIIT-UNAM grant no. IV 100220 (proyecto especial COVID-19) and IN115720.

Acknowledgements. We thank the Mexican Federal Health Secretary and the WHO Strategic Advisory Group of Experts (SAGE) on immunization working group on COVID-19 vaccines for facilitating relevant data and policy recommendations.

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
