## [Peer Review File · Royal Society Open Science]

Review History

RSOS-202240.R0 (Original submission)

Review form: Reviewer 1

Is the manuscript scientifically sound in its present form?

Yes

Are the interpretations and conclusions justified by the results?

Yes

Is the language acceptable?

Yes

Do you have any ethical concerns with this paper?

No

Have you any concerns about statistical analyses in this paper?

No

Recommendation?

Accept with minor revision (please list in comments)

Comments to the Author(s)

The sentence from lines 23-27 is too long. I would break it into two sentences by erasing but and use however as a connector. The phrase "in developing countries" sounds restrictive. You can use "across all countries" or use another phrase.

Review form: Reviewer 2

Is the manuscript scientifically sound in its present form?

No

Are the interpretations and conclusions justified by the results?

Yes

Is the language acceptable?

Yes

Do you have any ethical concerns with this paper?

No

Have you any concerns about statistical analyses in this paper?

No

Recommendation?

Major revision is needed (please make suggestions in comments)

Comments to the Author(s)

Review comments for "The trade-off between mobility and vaccination for COVID-19 control: a metapopulation modeling approach" by Saldaña and Velasco-Hernández. A metapopulation risk structured model is developed and used to study the impact of vaccines on the spread of COVID-19 compared to mobility restrictions. This is an interesting study, but I believe it is too theoretical and not associated with actual COVID-19 data from any specific country or region. This weakens the biology and limits readership from the general scientific community. Some specific comments are presented below.

No pre-symptomatic class is included in the model, although this class has been shown to contribute significantly to disease transmission.

The authors use latent period and incubation period interchangeably in this manuscript. But these are two different periods. Also, is 2-14 days the range of the incubation period or the mean of the incubation period?

The equation for the vaccinated exposed class is wrong. See the removal terms.

Computation of the basic reproduction number in the presence of movement is not clear. After ordering the subsystem with infections, one expects to have 9×9 and not 3×3 matrices. The authors did not clarify why they had 3×3 matrices instead. Also, I don't think that justifying not including the class reported cases because it does not contribute to the force of infection is reasonable. It is not also clear why the next generation operator approach for the effective reproduction number is 4×4 .

Some of the highly effective vaccines against COVID-19 like the Moderna and Pfizer vaccines require two doses to achieve optimal protection. But unfortunately, this manuscript is silent about this fact. It is important to consider the second dose.

Decision letter (RSOS-202240.R0)

Dear Dr Saldaña

On behalf of the Editors, we are pleased to inform you that your Manuscript RSOS-202240 "The trade-off between mobility and vaccination for COVID-19 control: a metapopulation modeling approach" has been accepted for publication in Royal Society Open Science subject to minor revision in accordance with the referees' reports. Please find the referees' comments along with any feedback from the Editors below my signature.

Please submit your revised manuscript and required files (see below) no later than 7 days from today's (ie 22-Apr-2021) date. Note: the ScholarOne system will 'lock' if submission of the revision is attempted 7 or more days after the deadline. If you do not think you will be able to meet this deadline please contact the editorial office immediately.

on behalf of Dr Pierre Magal (Associate Editor) and Glenn Webb (Subject Editor)
openscience@royalsociety.org

Associate Editor Comments to Author (Dr Pierre Magal):

I don't see any major problem in both report, just some corrections are needed. So please follow the referee comments and I think the paper should be accepted with minor revisions.

Reviewer comments to Author:
Reviewer: 1

Comments to the Author(s)

The sentence from lines 23-27 is too long. I would break it into two sentences by erasing but and use however as a connector. The phrase "in developing countries" sounds restrictive. You can use "across all countries" or use another phrase.

Reviewer: 2

Comments to the Author(s)

Review comments for “The trade-off between mobility and vaccination for COVID-19 control: a metapopulation modeling approach” by Saldaña and Velasco-Hernández. A metapopulation risk structured model is developed and used to study the impact of vaccines on the spread of COVID-19 compared to mobility restrictions. This is an interesting study, but I believe it is too theoretical and not associated with actual COVID-19 data from any specific country or region. This weakens the biology and limits readership from the general scientific community. Some specific comments are presented below.

No pre-symptomatic class is included in the model, although this class has been shown to contribute significantly to disease transmission.

The authors use latent period and incubation period interchangeably in this manuscript. But these are two different periods. Also, is 2-14 days the range of the incubation period or the mean of the incubation period?

The equation for the vaccinated exposed class is wrong. See the removal terms.

Computation of the basic reproduction number in the presence of movement is not clear. After ordering the subsystem with infections, one expects to have 9×9 and not 3×3 matrices. The authors did not clarify why they had 3×3 matrices instead. Also, I don't think that justifying not including the class reported cases because it does not contribute to the force of infection is reasonable. It is not also clear why the next generation operator approach for the effective reproduction number is 4×4 .

Some of the highly effective vaccines against COVID-19 like the Moderna and Pfizer vaccines require two doses to achieve optimal protection. But unfortunately, this manuscript is silent about this fact. It is important to consider the second dose.

===PREPARING YOUR MANUSCRIPT===

===PREPARING YOUR REVISION IN SCHOLARONE===

Author's Response to Decision Letter for (RSOS-202240.R0)

See Appendix A.

Decision letter (RSOS-202240.R1)

Dear Dr Saldaña,

I am pleased to inform you that your manuscript entitled "The trade-off between mobility and vaccination for COVID-19 control: a metapopulation modeling approach" is now accepted for publication in Royal Society Open Science.

COVID-19 rapid publication process:

We are taking steps to expedite the publication of research relevant to the pandemic. If you wish, you can opt to have your paper published as soon as it is ready, rather than waiting for it to be published the scheduled Wednesday.

This means your paper will not be included in the weekly media round-up which the Society sends to journalists ahead of publication. However, it will still appear in the COVID-19

Publishing Collection which journalists will be directed to each week (<https://royalsocietypublishing.org/topic/special-collections/novel-coronavirus-outbreak>).

If you wish to have your paper considered for immediate publication, or to discuss further, please notify openscience_proofs@royalsociety.org and press@royalsociety.org when you respond to this email.

on behalf of Dr Pierre Magal (Associate Editor) and Glenn Webb (Subject Editor)
openscience@royalsociety.org

Appendix A

April 26, 2021

Dr. Pierre Magal and Dr. Glenn Webb

Editors

Subject: Revision of Manuscript ID RSOS-202240 "The trade-off between mobility and vaccination for COVID-19 control: a metapopulation modeling approach".

Dear professors Magal and Webb,

Thank you for your email dated April 22, 2021. We thank the reviewers' comments on our manuscript.

We have carefully reviewed the comments and have modified the manuscript accordingly. Our responses are given in a point-by-point manner below. The corresponding changes are highlighted (in red) in the manuscript. We hope that the revised version is now suitable for publication and look forward to hearing from you.

Thank you once again for the invitation to contribute to this volume.

Yours sincerely,

Fernando Saldaña and Jorge X. Velasco-Hernández.

Response to reviewer 1:

Thank you very much for your review of our paper.

- The sentence from lines 23-27 is too long. I would break it into two sentences by erasing but and use however as a connector. The phrase "in developing countries" sounds restrictive. You can use "across all countries" or use another phrase.
- **We have corrected this issue exactly as suggested.**

Response to reviewer 2:

Thank you very much for your review of our paper. We have answered each of your points below.

- No pre-symptomatic class is included in the model, although this class has been shown to contribute significantly to disease transmission.
- **We agree with the reviewer. Our rationale for not including this compartment in our model is that the contribution of the pre-symptomatic class to SARS-CoV-2 spread can be considered as part of the asymptomatic class, particularly because our compartment E is of exposed individuals that remain in E, on average, the length of the incubation period. Moreover, taking into account the lack of quality data in such classes in the Mexican context, we believe such simplification is appropriate. Nevertheless, we now mention this limitation in the Discussion section (see the second paragraph).**
- The authors use latent period and incubation period interchangeably in this manuscript. But these are two different periods. Also, is 2-14 days the range of the incubation period or the mean of the incubation period?
- **We agree with the reviewer's commentary. The existence of a pre-symptomatic infectious period implies that the latent period is different from the incubation period (the incubation period is longer than the latent period). We indeed used such terms interchangeably (In the paragraph after the model equations we wrote that $1/k_i$ is the mean latent period. However, in subsection 2 (c), we wrote mean incubation period). We have corrected this inconsistency in the text. The incubation period of COVID-19 is on average 5-7 days but can be as long as 14 days (so the range is 2-14 days and the mean 5-7 days). We have corrected such sentences in subsection 2 (c). Thank you for your observation.**
- The equation for the vaccinated exposed class is wrong. See the removal terms.
- **Thanks for noticing such typos. We have corrected the equation.**

- Computation of the basic reproduction number in the presence of movement is not clear. After ordering the subsystem with infections, one expects to have 9 x 9 and not 3 by 3 matrices. The authors did not clarify why they had 3 by 3 matrices instead. Also, I don't think that justifying not including the class reported cases because it does not contribute to the force of infection is reasonable. It is not also clear why the next generation operator approach for the effective reproduction number is 4 by 4.
- **As the reviewer mentioned, in the computation of R0 in the presence of movement, one expects to have F, V matrices with 9 x 9 dimension. Based on the equations (2.4)-(2.5), we believe the reviewer was confused because such matrices have dimension 3 x 3. Nevertheless, such 3 x 3 matrices are only the components of the 9 x 9 matrices that form the next generation matrix (see this line before equation (2.4): we obtain a block matrix $F=[\tilde{F}^{ij}]$, where for $i,j=1,2,3$, \tilde{F}^{ij} is an 3 x 3 matrix). The V matrix is defined in a similar manner, so the dimensions of such matrices are 9 x 9. In the same manner, the next generation matrix for the effective reproduction number has dimension 12 by 12 because the vaccinated exposed class for each patch should be considered in addition to the other classes. Finally, since the reported class does not contribute to the force of infection, considering such class will result in an extra row and column for each \tilde{F}^{ij} matrix (see equation (2.4)) full of zeros. Hence, the spectral radius of the next-generation matrix is not affected by such additions.**
- Some of the highly effective vaccines against COVID-19 like the Moderna and Pfizer vaccines require two doses to achieve optimal protection. But unfortunately, this manuscript is silent about this fact. It is important to consider the second dose.
- **Thank you for the commentary. As mentioned in the introduction, rather than obtaining quantitative predictions of the epidemic course, we explored a spectrum of possibilities to gain qualitative insight on the population-level impact of vaccine introduction. So, for our objectives, we assume that the vaccinated class approximates a vaccination program in which the 2 shots needed for full protection are already given. We mentioned this limitation of our model in the discussion section (second paragraph).**